# Phytochrome-Interacting Proteins

**DOI:** 10.3390/biom14010009

**Published:** 2023-12-21

**Authors:** Gero Kaeser, Norbert Krauß, Clare Roughan, Luisa Sauthof, Patrick Scheerer, Tilman Lamparter

**Affiliations:** 1Karlsruhe Institute of Technology (KIT), Joseph Gottlieb Kölreuter Institut für Pflanzenwissenschaften (JKIP), Fritz-Haber-Weg 4, D-76131 Karlsruhe, Germany; eg5254@kit.edu (G.K.); norbert.krauss@kit.edu (N.K.); clare.roughan@student.kit.edu (C.R.); 2Charité—Universitätsmedizin Berlin, Corporate Member of Freie Universität Berlin and Humboldt-Universität zu Berlin, Institute of Medical Physics and Biophysics, Group Structural Biology of Cellular Signaling, Charitéplatz 1, D-10117 Berlin, Germany; luisa.herder@charite.de (L.S.); patrick.scheerer@charite.de (P.S.)

**Keywords:** PIF3, PKS2, Cry, plant, bacterial, fungal, interaction

## Abstract

Phytochromes are photoreceptors of plants, fungi, slime molds bacteria and heterokonts. These biliproteins sense red and far-red light and undergo light-induced changes between the two spectral forms, Pr and Pfr. Photoconversion triggered by light induces conformational changes in the bilin chromophore around the ring C-D-connecting methine bridge and is followed by conformational changes in the protein. For plant phytochromes, multiple phytochrome interacting proteins that mediate signal transduction, nuclear translocation or protein degradation have been identified. Few interacting proteins are known as bacterial or fungal phytochromes. Here, we describe how the interacting partners were identified, what is known about the different interactions and in which context of signal transduction these interactions are to be seen. The three-dimensional arrangement of these interacting partners is not known. Using an artificial intelligence system-based modeling software, a few predicted and modulated examples of interactions of bacterial phytochromes with their interaction partners are interpreted.

## 1. Introduction

Phytochromes are photoreceptor proteins that regulate multiple light effects in plants, bacteria, fungi and heterokonts [1,2,3,4]. The protein binds a bilin chromophore covalently to a cysteine residue. As in any photoreceptor, the chromophore absorbs light and this triggers protein conformational changes that initiate a signal transduction cascade. Phytochromes have a special feature, photoreversibility, which describes the switching between two spectrally different forms, Pr and Pfr, by light. This conversion is triggered by an isomerization of the chromophore. Most members of the large group of phytochromes are synthesized in the Pr form, while Pfr can only be formed by light. To achieve conversion from Pfr to Pr, there are two possibilities, either via photoconversion or via dark reversion. Few phytochromes are comparable to this group but have a stable Pfr, i.e., the conversion from Pr to Pfr is only possible by light. A third group, the so-called bathy phytochromes, have a Pfr dark form [5,6]. They are also synthesized in the Pr form, but by dark conversion, Pr converts to Pfr. Pr can then only be formed by light. Three different bilins, phytochromobilin, phycocyanobilin and biliverdin, can be selected by phytochromes (Figure 1). The choice depends on the species, i.e., which chromophore is synthesized, and on the position of the chromophore binding cysteine [7]. There are also few examples of other chromophores with still different spectral properties [8], and the group of cyanobacteriochromes [9], biliproteins that probably evolved out of phytochromes and are absorbed in various different spectral ranges.

Phytochromes are multidomain proteins. The canonical phytochromes have a PAS (period, Arndt, single-minded)-GAF (cGMP-specific phosphodiesterases, adenylyl cyclases and FhlA)-PHY (phytochrome) tridomain in the N-terminus of the protein (Figure 2). The GAF domain gives rise to the cyanobacteriochromes, in which GAF is combined with different other domains. Cyanobacterial proteins, in which GAF and PHY are present but a PAS domain is missing, are termed knotless phytochromes [8]. Our focus here is on the canonical phytochromes. In the C-terminus, most phytochromes have a histidine kinase or a histidine kinase-like domain. Plant phytochromes are comparable with the typical bacterial phytochromes but carry two additional PAS domains between the PHY domain and the histidine kinase-like module (Figure 2). However, plant phytochromes lose their histidine kinase activity. 

All phytochromes analyzed so far have a dimer arrangement with usually two identical subunits, although heterodimer formation is described for few plant phytochromes [13]. As far as recent structure determinations are concerned, subunits of bacterial phytochromes are arranged in a parallel manner (Figure 3). Surprisingly, plant phytochrome structures have parallel histidine kinase-like regions, but the photosensory core modules (PCM modules) are arranged in an antiparallel manner, and the overall dimer forms an unexpected, complex and asymmetric arrangement (Figure 3).

Phytochromes were discovered as the second group of photoreceptors after the opsins. Phytochrome discovery is based on action spectroscopy of plant effects such as flower induction [17], seed germination [18] or de-etiolation [19], followed by spectral detection which is based on the Pr-Pfr photoreversibility [20]. In fact, phytochromes control a large number of developmental effects in plants and can therefore be regarded as the most relevant plant photoreceptors [21]. The later discoveries of phytochromes in bacteria, fungi, slime molds and heterokonts were based on genome sequences, and consequently, the known light effects as controlled by phytochrome are rare in these groups: the organisms are much more diverse, and it is expected that the phytochrome-controlled effects are also diverse, i.e., different from one bacterium to another. Because light effects are hard to recognize in bacteria, they may remain hidden until genes are modified and responses are targeted by researchers. In cyanobacteria, the cyanobacteriochromes control the effects of chromatic adaptation [22] or phototaxis [23], whereas the only effect known for a canonical phytochrome is an effect on biofilm formation, which is probably not light dependent [24]. In the photosynthetic bacteria *Bradyrhizobium* and *Rhodopseudomonas*, the synthesis of photosynthesis proteins and bacteriochlorophyll are under the phytochrome’s control [5]. In the soil bacterium *Agrobacterium fabrum*, the gene transfer to plants and the plasmid transfer to other bacteria by conjugation are under the control of phytochrome [25,26,27]. 

The signal transduction pathway starts as follows: Light is absorbed by the bilin chromophore, which undergoes an isomerization around the ring C-D-connecting methine bridge from the *Z* to the *E* configuration as the first step of the photocycle [28,29]. The rotation of the ring D and its differential interaction with binding amino acids results in conformational changes in the protein. These are translated into modulations of the histidine kinase and changes in its enzyme activities, in other enzymatic activities or in interactions with other proteins as the next step of signal transduction. The three-dimensional structures of phytochrome’s PAS-GAF-PHY tridomains are well known [16,30,31]. The GAF domain forms a chromophore pocket, PAS and PHY domains also interact with the chromophore and an exceptional knot is formed between PAS and GAF domains. Upon Pr to Pfr conversion, the secondary structure of the so-called tongue of the PHY domain changes from ß-sheet to α-helix, and this step might initiate protein changes in the C-terminus. Meanwhile, also crystal structures of full-length phytochromes are known [32,33], and due to the successful cryogenic electron microscopy (cryo-EM) structure determination, a few 3D structures of full-length plant phytochromes are also known [34,35,36]. Phytochromes form homodimers, and the PAS-GAF-PHY tridomains are usually arranged in a parallel manner, although antiparallel arrangements have been described for the cyanobacterial phytochrome Cph1 and the *A. fabrum* phytochrome Agp1 [16,31]. The cryo-EM structure of plant phytochrome B determined recently shows an asymmetric and quite unexpected arrangement of subunits with antiparallel PAS-GAF-PHY tridomains, whereas the full-length structures of DrBphP and *Ipomea* phytochromes showed parallel arrangements of both subunits [37,38]. 

In this overview, we summarize the interactions of phytochromes with other proteins. The interactions are relevant for the first steps of signal transduction or for the intracellular migration of phytochromes. Most interacting proteins are known for plant phytochromes, which is due to the long research period since the discovery of plant phytochromes, the large number of research groups interested in plant phytochromes and the higher complexity of plant cells compared to bacteria.

## 2. Interaction Partners of Plant Phytochromes

Most phytochrome interaction proteins have been found for the plant *Arabidopsis thaliana*. This model plant has five phytochromes. Interacting proteins are listed in the Biogrid database [39]. The entries for PhyA are, for example, found here: https://thebiogrid.org/22724/table/arabidopsis-thaliana/PhyA.html. Our selection below covers most of the database entries. In most cases, PhyA and/or PhyB are the interaction partners, and the experiments are usually carried out with the model plant *Arabidopsis thaliana*.

## 3. Homodimer and Heterodimer Formation

Phytochromes are dimeric proteins. The interaction between monomers may be regarded as a very stable protein interaction of phytochromes. Dimerization has been long documented and analyzed [40,41,42]. Recently, it has been discovered that the arrangement of subunits in PhyA and PhyB is significantly different from what had been expected. The arrangement is asymmetric, and the subunits are aligned parallel to each other in the C-terminal part, the histidine kinase-like region, and antiparallel to each other in the N-terminal PCM. This asymmetric arrangement poses new questions about the interaction with other proteins. Is it a specific interaction with one subunit, is it an interaction with either subunit of both sides, or are both subunits required?

Typically, both subunits of the dimer have identical sequences. However, since there are different phytochromes in a species, there is the possibility that subunits of different phytochromes form a dimer, although a mixed formation would not necessarily turn out from standard biochemical assays such as Western blotting. Indeed, co-precipitation of Myc-tagged phytochromes showed that heterodimers can be formed between all type II phytochromes of Arabidopsis (PhyB, PhyC, PhyD and PhyE), whereas PhyA seems to be excluded from mixed dimer formation [13].

## 4. Interaction with PIF3 and Other PIF Proteins

The interaction between phytochromes and phytochrome interacting factor 3 (PIF3) was the first interaction for a phytochrome with another protein to be discovered. This is probably the best-studied interaction of a phytochrome with another protein, as seen by the literature statistics: in *Web of Science* (a paid-access platform for reference and citation data from academic journals), there are 420 articles and reviews on PIF3 [43,44,45,46]. PIF3 was found by a yeast two-hybrid screening using an Arabidopsis cDNA library as prey and PhyB as bait [47]. Considering the evidence indicating that the “signaling” C-terminal domain of the phytochromes might be directly involved in the interaction with signaling partners [48], Ni et al. used the non-photoactive C-terminal PAS-PAS-HK (see Figure 2) of PhyB as bait in the yeast two-hybrid system.

To investigate the functional relevance to phytochrome signaling, transgenic Arabidopsis seedlings expressing both sense and antisense PIF3 constructs were examined, evaluating their impact on photoresponsiveness under constant red light or constant far-red light conditions, which are indicative of PhyB and PhyA activity, respectively. Transgenic Arabidopsis seedlings, which had reduced levels of PIF3 through antisense techniques, showed significantly decreased responsiveness to light signals detected by either PhyB or PhyA. Thus, PIF3 is functionally involved in both the PhyA and PhyB signaling pathways within plant cells, which aligns with its ability to bind to both photoreceptors [49].

Following the discovery of the interaction between PIF3 with the C-terminal domain of PhyB, an in vitro pull-down assay was performed to verify the interaction with the full-length, photoactive PhyB holoprotein. In their study, Ni et al. [50] used PIF3 immobilized on beads as bait and full-length phytochrome molecules carrying a chromophore as prey. The phytochrome molecules were synthesized in vitro [50]. The results demonstrated that PhyB binds to PIF3 only upon light-induced conversion to its biologically active Pfr form. When PhyB reverted to its inactive Pr form, PIF3 dissociated from the phytochrome. This indicates that PhyB signaling relies on the specific recognition of PIF3 by the active Pfr form of the photoreceptor. While initial experiments indicated an interaction between PIF3 and the C-terminal region of the phytochrome, it became evident that a phytochrome lacking a C-terminus can initiate light-induced signal transduction, and in fact, it can do so even more efficiently than the full-length protein [51]. The C-terminus might function in a modulatory way or could initiate pathways that differ from the standard signaling pathways, providing an explanation for PIF3’s interaction with the C-terminus. Indeed, the Ni et al. experiments (1999) [50] showed that there is a Pfr-dependent interaction of PIF3 with the N-terminal PAS-GAF-PHY domain of the phytochrome. 

In a comprehensive mutagenesis-based “yeast reverse-hybrid screen”, Kikis et al. identified four amino acids of PhyB that are necessary for PIF3 binding [52]. The same PhyB mutants were also reduced in their light responses in Arabidopsis. All four amino acids clustered in the knot region of PhyB, which is formed by PAS and GAF domains in the N-terminus of the protein. These experiments gave deep insight into the spatial interactions between PhyB and PIF3. Because the C-terminus of PhyB is too distant from the knot region, it seems unlikely that PIF3 interacts with the C-terminus and the N-terminal knot at the same time. 

In 2006, Al Sady et al. [53] developed antibodies targeting the low-abundance native PIF3 protein. They also created various transgenic Arabidopsis lines that expressed different forms of PIF3 fusion proteins labeled with fluorescent markers and epitope tags. Through Western blot analysis, they confirmed and expanded the data of Bauer et al. [54], who demonstrated that the immunochemically measurable native PIF3 protein decreases below detectable levels in wild-type Arabidopsis seedlings within 260 min of exposure to continuous red light. These studies showed that the interaction with Phy induces the degradation of PIF3. 

PIF3 is a member of the basic helix-loop-helix (bHLH) family and localized in the nucleus [47,50]. It functions as a negative regulator of photomorphogenesis by repressing the expression of genes involved, like those associated with chlorophyll biosynthesis and light-induced growth processes [52,55]. Under dark conditions or in the presence of inactive phytochromes, PIF3 is stable and able to bind to specific DNA sequences in the promoters of target genes, thereby repressing their expression. In the light, plant phytochrome in its active Pfr conformation is quickly translocated into the nucleus. These active phytochromes can interact with PIF3 and induce its degradation by phosphorylation [56] and sumoylation [57] or prevent its binding to DNA by competing with the DNA binding. The degradation or inhibition of PIF3 by active phytochromes allows for the activation of photomorphogenesis-related genes and promotes plant growth and development [53,58,59]. 

In Arabidopsis, there are six other proteins that are homologous to PIF3, namely, PIF1, PIF4 [60], PIF5 [61], PIF6 [62], PIF7 [63] and PIF8 [60]. Homologous PIFs have also been examined in other plants [64]. All members seem to bind phytochrome specifically in the Pfr form, but the DNA binding positions differ. The variability in PIFs is thus one way to allow for multiple variations in light control via phytochrome. 

The signal transduction cascade for PIF3 performs as follows: Upon irradiation of the plant and photoconversion from Pr to Pfr, the phytochrome transfers from the cytosol to the nucleus. Here, it interacts with PIF3 in a Pfr specific way. This leads to the phosphorylation of PIF3 through a kinase which also interacts with the phytochrome and PIF3. Upon sumoylation, PIF3 is targeted for degradation. Because, in general, PIF3 acts as a negative transcriptional regulator, the degradation of this protein leads to an induction of expression in the light (see model in Figure 4). Due to its early discovery, PIF3 is the best-analyzed member of the PIF family but probably also the most relevant with respect to light regulation of plant responses. 

## 5. Interaction with Cryptochrome (Cry)

Cryptochromes are widely distributed flavoprotein blue light photoreceptors that are homologous to the photolyases, which are enzymes that repair DNA damage. Plants typically contain two to three cryptochromes and two other flavoprotein photoreceptors, the phototropins. Already, in the early days of phytochrome research [65,66], it was found that many phytochrome effects can also be induced by blue light. Because the bilin chromophore of the phytochrome absorbs not only red light but to a lesser extent also blue light, both light qualities could equally induce responses via the phytochrome. However, separate blue light photoreceptors could as also cause the blue light effects. The gene of cryptochrome was discovered in 1994 [67] and the gene of phototropin was discovered in 1997 [68].

An interaction between Cry1 and plant phytochrome A was first demonstrated by the phosphorylation of Cry1 by purified PhyA, and by yeast two-hybrid techniques [69]. The interaction seems independent of the photochromic state (Pr or Pfr) of the phytochrome, because phosphorylation of Cry by the phytochrome was independent of the respective form of the latter (but see [70]). 

While the interaction between Phy and PIF3 results in a clear action, i.e., the degradation of PIF3, the result of the phytochrome/cryptochrome interaction is open. However, studies on the coaction of both photoreceptors have continued [70,71].

## 6. Interaction with ARR4

Bacterial and fungal phytochromes often have histidine kinase domains and function as light-regulated histidine kinases that phosphorylate cognate response regulator proteins. Plant phytochromes have a module at the C-terminus which has evolved from histidine kinases but has no phosphorylating function and the regions do not dimerize in the way characteristic for histidine kinases [35]. Nevertheless, the attention has been drawn to Arabidopsis homologs for the *E. coli* CheY response regulator. Mutant studies have already identified response regulator sequences in cytokinin signal transduction [72]. One of these, ARR4 (Arabidopsis response regulator 4), was found to be involved in light signal transduction. Subsequent in vitro studies demonstrated that ARR4 interacts with the N-terminus of PhyB, but not of PhyA [73]. ARR4 knockout mutants show reduced sensitivity to red light. One action of ARR4 is that it modulates dark reversion of PhyB. 

## 7. Interaction with Phototropin

While interactions with cryptochrome and PIFs probably take place in the nucleus of a plant cell, an interaction between the phytochrome and phototropin at the plasma membrane has been shown for the moss *Physcomitrium patens* [74]. The tip cell of moss filaments grows phototropically toward the light. This response is controlled by the phytochrome (and possibly phototropin). A gradient of activated photoreceptor can only be formed by unilateral light if the photoreceptors are immobilized, most likely at the plasma membrane. The interaction between the phytochrome and phototropin at the plasma membrane could fulfil both functions. 

## 8. Interaction with COP1

Constitutive photomorphogenesis 1 (COP1) is a ubiquitin ligase that is involved in protein degradation. Dark grown COP1 and DET (de-etiolated) mutant seedlings are similar in many aspects to wild-types grown in the light. The interaction of COP1 with the N-terminal of PhyB in vitro has been shown by pull-down assays [75]. COP1 is involved in many photomorphogenic responses, probably by inducing degradation as part of plant proteasomes. The light-induced degradation of plant PhyA was found already very early in phytochrome research. Here, PhyB is thought to be degraded through the COP1 proteasome in the nucleus. 

## 9. Interaction with NDPK2

In a yeast two-hybrid screen using the C-terminal part of PhyA as a bait, the nucleoside diphosphate kinase 2 (NDPK2) was isolated [76]. In vitro characterization showed that the interaction takes place between one PAS domain A of the C-terminus of PhyA and the C-terminal part of NDPK2 [77]. The enzyme catalyzes the phosphotransfer of the gamma phosphate of ATP to other nucleoside diphosphates and thus plays an important role in nucleotide metabolism. NDPK2 was shown to be involved in phytochrome photoregulation, e.g., of starch formation [77,78,79,80]. 

An open question in NDPK2 function is its localization to the chloroplast [80] via a chloroplast target sequence. Plant phytochromes reside in the cytoplasm, at the plasmalemma or in the nucleus, but have never been reported to be in the chloroplast. The interaction between NDPK2 and the phytochrome is thus either not relevant for in vivo functions or is relevant for the protein on its way to the chloroplast.

## 10. Interaction with PKS1

PKS1 (PHYTOCHROME KINASE SUBSTRATE 1) is a protein in *Arabidopsis thaliana* that has been identified in a yeast two-hybrid screen with C-terminal PhyA and PhyB as bait. The interaction in vitro with the full-length GST-tagged PhyA and PhyB has been demonstrated: the interaction appears to be independent from Pr or Pfr [81]. A homolog PKS2 fulfils similar functions [82]. PKS1 is phosphorylated by phytochromes in response to light, and this phosphorylation is thought to activate downstream signaling components. In addition to its role as a substrate of phytochrome-mediated signaling, PKS1 has also been shown to interact with other proteins involved in light signaling pathways, such as the photoreceptors cryptochrome or phytotropin. This suggests that PKS1 may play a broader role in regulating plant responses to light beyond its interactions with phytochromes.

## 11. Interaction with SPA1

The suppressor of phytochrome A (SPA1) has been shown to interact with phytochromes by using the yeast two-hybrid system and by using FRET-based methods in plant cells [83]. SPA1 is involved in the regulation of photoperiodic flowering and regulates circadian rhythms. It is required for the suppression of photomorphogenesis in dark-grown seedlings and for normal elongation growth of adult plants. As an integral component of the COP1-SPA-E3 ubiquitin–protein ligase complex, it is involved in degradation of PhyA as well as in HY5, HFR1, LAF1 and CO degradation.

## 12. Other Plant Proteins Involved in Phy Signal Transduction

Based on mutant results and physiological investigations of light effects in Arabidopsis, genes and proteins were identified that play a central role in phytochrome signal transduction, but for which no direct interaction with phytochromes has been found. These are, for example, HY5 (long hypocotyl 5), a transcription factor [84,85], HFR1, a bHLH transcription factor [86], or DET1 (DEETIOLATED 1). All genes arose from mutant studies with Arabidopsis.

## 13. Interaction with FHY1 and FAR1

FAR-RED ELONGATED HYPOCOTYL 1 (FHY1) and FAR-RED IMPAIRED RESPONSE 1 (FAR1) are proteins that are involved in the translocation of PhyA into the nucleus. In dark-grown plant tissue, the phytochrome (in the Pr form) is localized in the cytoplasm, but upon light-induced conversion into Pfr, the protein is translocated into the nucleus, although a subfraction remains in the cytosol or bound to the plasma membrane. The interaction between PhyA and FHY1 or FAR1 [87,88] is dependent on Pfr. In the nucleus, the interaction with, e.g., PIF proteins mediates transcriptional regulation, and the COP1 interaction mediates the degradation of phytochromes. During the transport of phytochromes, the partner proteins cycle between the nucleus and cytoplasm (see the model in Figure 4). During constant irradiation, phytochromes cycle between Pr and Pfr because light is absorbed by both forms. [89]. This explains the switch from binding to non-binding. Nuclear transport, Pr-Pfr cycling and Phy degradation can explain the shift of action spectrum toward far-red. 

## 14. Fungi

The first fungal phytochrome was discovered in 2004 [90]. Two proteins were shown to physically interact with phytochrome FphA from *Aspergillus* (or *Emericella*) *nidulans*, VeA and LreB [91]. It was shown that FphA, LreA, LreB and VeA form a regulatory complex in vivo. VeA is a transcription factor with a nuclear localization sequence that controls the transcription of hundreds of genes. LreB, which is homologous to WC2 of *Neurospora crassa*, is also a transcription factor which also interacts with the WC1 homolog and blue light receptor LreA. It was shown by bimolecular complementation that LreB interacts with the C-terminus of FphA (histidine kinase + response regulator), whereas VeA interacts with a part of the histidine kinase.

## 15. Bacteria

The first bacterial phytochromes were found in 1996 or 1997 [22,92,93], also in cyanobacteria. Due to the many available genome sequences, it is known that phytochromes are broadly distributed among bacterial phyla but are not present in archaebacteria. The broad diversity of bacterial sequences as compared to eukaryotes is also reflected in the high diversity of bacterial phytochrome sequences. Despite this diversity, little is known about the biological functions of bacterial phytochromes. The first clear example of a bacterial phytochrome effect was the regulation of photosynthesis in the proteobacterium *Bradyrhizobium* [5]. In *Agrobacterium fabrum,* it is known that conjugation and gene transfer to plants is controlled by phytochromes [27,94]. 

Since bacterial phytochromes were discovered long after plant phytochromes, and research on these phytochromes focuses on the photoreceptor’s biochemical studies, the list of confirmed interaction partners is more limited.

## 16. Interactions of Bacterial Phytochromes with Response Regulators

The majority of bacterial phytochromes have a C-terminal response histidine kinase region, and often a response regulator is encoded by a neighboring gene. The canonical mechanism, as exemplified for the first discovered bacterial phytochrome Cph1 [92], is that the phytochrome histidine kinase autophosphorylates in a Pr/Pfr-dependent manner (Pr strong and Pfr weak, see also [95,96]) and that the phosphate is transferred to the response regulator. This phosphotransfer mechanism follows the typical scheme for other histidine kinases [97]. Several bacterial phytochrome systems differ from the classical pattern. In two cases, DrBphP [98] and Agp2 [99], no autophosphorylation was found. A phosphatase activity was described for DrBphP. In *Pseudomonas syringae* phytochrome, the kinase follows a low-Pr high-Pfr pattern [98]. In many bacterial phytochromes, such as Agp2 (and all fungal phytochromes), the response regulator is linked to the C-terminal part of the histidine kinase region (see also Figure 2). The biochemistry of histidine kinase—response regulator transphosphorylation—is yet only scarcely studied but both proteins do interact during the time of transphosphorylation. A structural study by Wahlgren et al. [37] shows the interaction of a response regulator and DrBphP at its C-terminus. We have modeled an interaction of a bacterial phytochrome with its cognate response regulator (see below). 

## 17. The Transcription Factor PpsR

The transcription factor PpsR regulates the formation of the photosynthetic apparatus in many bacteria. In *Rhodobacter sphaeroides* and other species, this regulation is a blue light effect, mediated via the BLUF protein AppA, which was shown to interact with PpsR in a light-dependent manner and in this way, reverse the repression of the inhibitory effect of PpsR on transcription [100]. In *Bradyrhizobium* and *Rhodopseudomonas* species, phytochromes play a major role in light regulation of photosystem formation [5]. The phytochrome reverses the repression of PpsR [101]. The interaction between the phytochrome and PpsR has been used for optogenetics [102]. 

## 18. Interaction between *Agrobacterium fabrum* Phytochromes Agp1 and Agp2

The soil bacterium *Agrobacterium fabrum* has two phytochromes, Agp1 and Agp2. Agp2 is a bathy phytochrome, i.e., adopts the Pfr form in darkness, whereas Agp1 belongs to the normal phytochromes with Pr ground state [25,26]. Both phytochromes have histidine kinase modules, and for Agp1, it was shown that autophosphorylation is high in the Pr and low in the Pfr form. *A. fabrum* uses two types of DNA transfer. During conjugation, plasmid DNA is transferred from one bacterium to another bacterium. The T-DNA of the Ti plasmid can be transferred to plant cells, thereby inducing tumor growth and opine production in the plant. It was found that both DNA transfer processes are light-dependent and controlled by phytochromes [27,94]. Mutant studies showed that in both cases, Agp1 and Agp2 act together in the regulation. It seemed therefore possible that both phytochromes interact. 

This interaction was indeed shown by different methods. When absorbance spectra were measured from Agp1 and Agp2 and from a mixed sample, the spectrum of the mixed sample differed from the sum of the spectra of single proteins. The rates of dark reversion were affected by the presence of the respective other phytochrome. The phosphorylation of Agp1 (in the Pr, but not in the Pfr form) was diminished by the presence of Agp2. Biliverdin assembly of Agp1 was significantly inhibited by the presence of Apg2. Agp1 and Agp2 were labeled with fluorophore and subjected to fluorescence measurements. The results clearly showed fluorescence resonance energy transfer [99]. All data showed that Agp1 and Agp2 do interact in solution. Ongoing measurements with fluorescence markers at different positions of Agp1 allowed conclusions to be made about the orientation of Agp1 and Agp2 to each other. 

## 19. Possible Interaction of Agp1 or Agp2 with TraA and VirD2

The impact of Agp1 and Agp2 on the conjugation process suggests interaction with conjugation proteins. It is unlikely that the regulation occurs via differential transcription, because microarray and proteome measurements showed no differences between the wild-type and the double knockout mutant or between light and dark [27,103]. *A. fabrum* has three TraA proteins that initiate conjugation. A TraA protein has several domains: a mob domain, which cleaves the DNA and forms a covalent DNA–protein adduct, a helicase domain, which unwinds double-stranded DNA, and a so-called BID domain [104]. TraA is the first protein in the chain of conjugation events and it is therefore tempting to assume that Agp1 or Agp2 directly interact with TraA. Our group is presently studying this putative interaction in vitro. The experiments are ongoing. We modeled possible interactions by Alphafold [15] and obtained reliable results that the TraA mob domain interacts with the histidine kinase of Agp1 (see below). 

The gene transfer from *A. fabrum* to plants is also regulated by light and by phytochromes, as noted above. The first protein in the cascade of DNA excision and transport is VirD2. The functions are like those of TraA discussed above. VirD2 cleaves T-DNA at the border sequences from the Ti plasmid, forms a covalent adduct with the DNA and migrates together with the DNA to the plant cell to be infected. The regulation of VirD2 expression by light or by phytochromes was not found in microarray and proteome studies. It was therefore suggested that Agp1 or Agp2 interact with VirD2 directly and thereby modulate its activity. Like TraA, VirD2 is the first protein in the DNA transfer cascade, and it is also the last.

Overall, while bacterial phytochromes are not as well studied as their plant counterparts, there is growing evidence to suggest that they play important roles in regulating bacterial physiology and behavior in response to light. Further research will be needed to fully understand the mechanisms underlying these responses and the signaling pathways involved.

## 20. Modeling the Interaction

To visualize the potential interactions of interacting proteins with phytochromes, we modeled these protein–protein interactions using the AI-based software Alphafold 2.3.1 (AF) [15]. This program predicts 3D structures of proteins even if no homologous structure is available. Interactions between different proteins can also be analyzed. In our approaches, the upper limit of molecular masses for the structures to be predicted was about 200 kDa. AF generates different models and provides quality criteria for each position in the modeled protein complex. It should be clearly emphasized here that these AF models have their limitations and they can never be perfect (also because of the different predictions). Such models need to be structurally and functionally tested by experimental approaches. We had the impression that a part of the AF models cannot be used, whereas some others can be used to generate a working hypothesis and, in this way, complement experimental approaches and increase the efficiency. 

We used PhyB as a protein–protein complex in combination with PIF3, PIF6, CRY1, or with zeitlupe [105], and PhyA in combination with NDPK2, as well as FphA in combination with VelA for AF structural modeling. In some cases, it was possible to model the potential interaction, but not for, e.g., PhyB and PIF6. Due to the 200 kDa limit, it was only possible to model phytochrome monomers in combination with other proteins, but it was not possible to model structures with a plant phytochrome dimer. Moreover, the overall folding of PhyA or PhyB differed from the cryo-EM structures published recently ([35] and PDB Codes: 7RZW, 8ISI, 8F5Z and 8ISJ). The cryo-EM structure shows an asymmetric and unexpected arrangement of the phytochrome domains, the subunits being arranged in parallel in the C-terminus and antiparallel in the N-terminus (Figure 3). The AF monomer model was not consistent with either monomer of the cryo-EM structure. Since the four published cryo-EM structures of full-length plant phytochromes match well with each other (RMSDs < 1.5), the overall fold of the AF prediction is certainly wrong. Another difficulty was that the AF models of PIF3 or PIF6 consisted mainly of loop structures with few α-helices and no ß-sheet regions, indicating difficult predictability. Other transcription factors have a high proportion of rigid secondary structures and only a few loop structures. The AF models for PIF3 or PIF6 were therefore probably also not correct. For these reasons, we thought that the AF models with the large plant Phy or the fungal Phy should not be discussed in more detail. We therefore focus on the shorter bacterial phytochromes for which we were able to calculate dimer models. The calculated overall fold of the bacterial phytochromes matched the parallel arrangement of the subunits in the cryo-EM structure of the recently published *Deinococcus* phytochrome [37]. Firstly, we calculated a dimer model for full-length Agp1 (Figure 3B and Figure 5A) and the corresponding response regulator (RR). The two subunits of the response regulator were located between the ATPase of the two Agp1 subunits, resulting in a four-leaf clover appearance when viewed from the C-terminal end of Agp1. The cryo-EM structure of the (engineered) full-length phytochrome of *Deinococcus* also contained a (by mutagenesis attachment) response regulator arranged in a similar manner, but the resolution was too low for a better comparison. For phosphotransfer from the Agp1 histidine kinase to the response regulator, the response regulator needs to come into the vicinity of His 528, the substrate for the kinase. For phosphorylation, the ATPase also must come into the vicinity of the substrate histidine. The proposed four-leaflet arrangement could allow both the ATPase and the RR to come into alternate contact with His 528. The AF folding model could contribute to a better understanding of the mechanism. Although the mechanisms of histidine kinase of other systems are largely known, we will not go into more detail here. 

The AF calculations with Agp1 and VirD2 or with Agp1 and TraA (mob domain) resulted in similar arrangements (Figure 5B,C). Both VirD2 and TraA (mob domain) were located between the two ATPase subunits. The overlay of all three modeled interactions is shown in Figure 5D. An interaction of Agp1 with TraA or with VirD2 has not yet been clearly demonstrated experimentally, but the impact of Agp1 and Agp2 on both gene transfer pathways makes it very likely that the AF models support the interaction. The fact that all three proteins were located at a nearly similar position of Agp1 could indicate an interesting mechanism: VirD2, TraA and RR could compete for the same binding site on Agp1 and thereby interfere with the phosphotransfer.

## 21. Conclusions

Signal transduction is mostly mediated by protein interactions, and numerous interaction partners have been identified for plant phytochromes. Research on bacterial and fungal phytochromes started later, and fewer interaction partners for phytochromes from these groups are known so far. This overview shows that there is no universal phytochrome interacting protein, and that the interactions fulfil different functions such as intracellular translocation, degradation or signal transduction. The interactions in the cell are certainly dynamic, so that interaction partners do not necessarily compete for the binding. However, if different proteins interact at similar positions on the phytochrome molecule, the binding competition could play a regulatory role. Knowing the nature of the interaction at the structural level would therefore be essential to better understand the mode of action and function. For PIF3, an interesting mass mutagenesis approach has shown that this protein interacts with the knot region of phytochromes, which is formed by the N-terminal PAS and GAF domains. 

It is also important to understand how the interaction dependent on Pr and Pfr is realized at the phytochrome level. Increasing structural details about phytochromes are emerging, and for some phytochromes (fragments), conformational changes in the protein from Pr and Pfr crystal structures are known, e.g., by time-resolved X-ray methods. In addition, cryo-EM structures have already been used to obtain some full-length phytochrome structures, and it is likely that phytochromes in combination with an interaction partner will be structurally analyzed primarily by cryo-EM in the future.

## Figures and Tables

**Figure 1 biomolecules-14-00009-f001:**
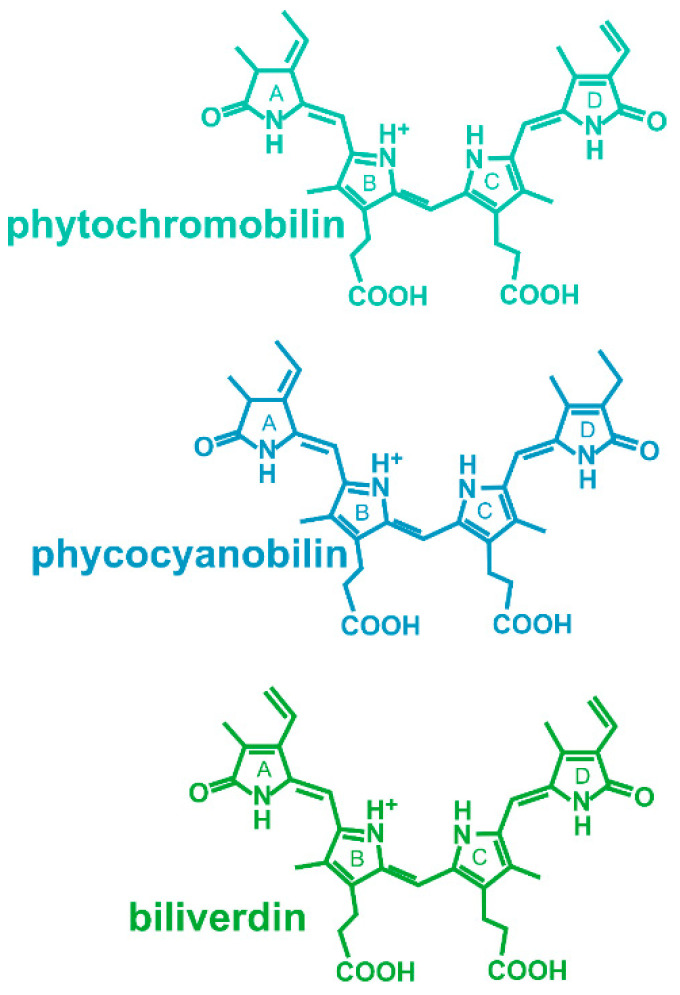
Chromophores of phytochromes. Phytochromobilin is the chromophore of plant phytochromes, phycocyanobilin is the chromophore of some cyanobacterial and green algal phytochromes and of cyanobacteriochromes, and biliverdin is the chromophore of bacterial and fungal phytochromes.

**Figure 2 biomolecules-14-00009-f002:**
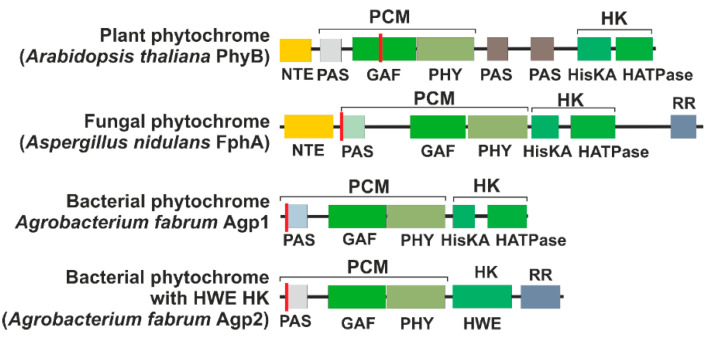
Domain arrangement of representative phytochromes. The red line indicates the position of the chromophore-binding cysteine. The abbreviations are NTE, N-terminal extension; PCM, photosensory core module; HK, histidine kinase (like) module; PAS, PAS domain [10]; GAF, GAF domain [11]; PHY, PHY domain [12]; HisKA, substrate and dimerization domain of histidine kinase; HATPase, ATPase domain of histidine kinase; HWE, HWE type of histidine kinase; RR, response regulator.

**Figure 3 biomolecules-14-00009-f003:**
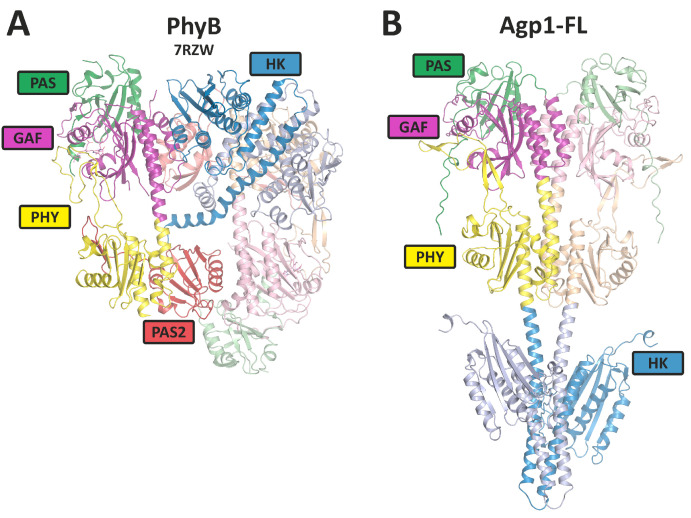
Overall rearrangement of the 3D structures of plant PhyB dimer and a phytochrome Agp1 dimer as representatives for plant and bacterial phytochromes. (**A**) The PhyB model is from cryo-EM studies (PDB [14] entry 7RZW); (**B**) the full-length Agp1 (Agp1-FL) model is based on Alphafold 2.3.1 [15] with 2 identical sequences (Uniprot-ID: Q7CY45) as entry (see also [16] for PCM structures). The domains are indicated by different colors. HK stands for histidine kinases (like).

**Figure 4 biomolecules-14-00009-f004:**
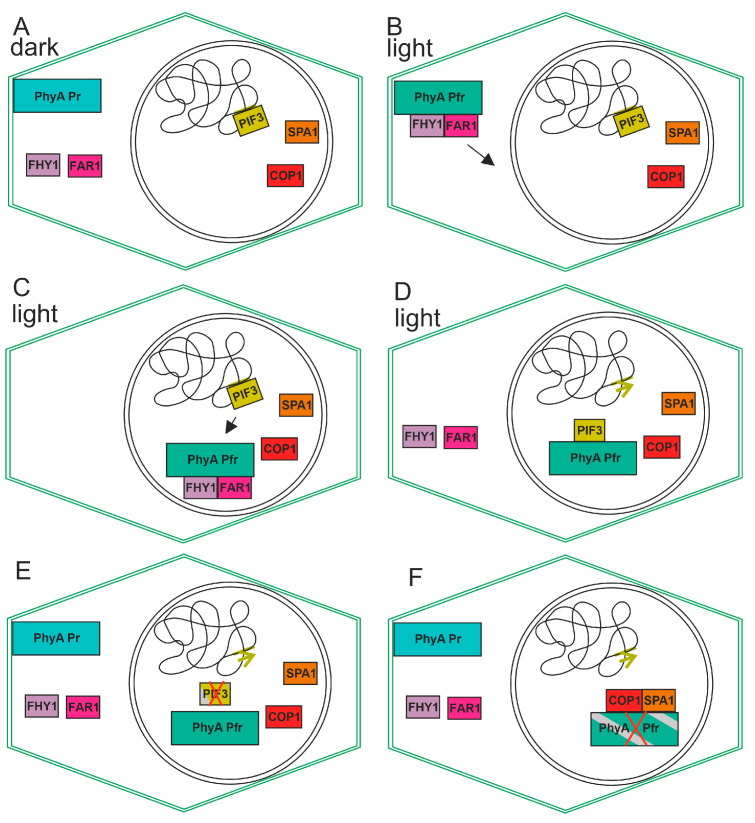
Cartoon for the action of plant phytochrome A. The cell borders are defined by the outer lines, while the nucleus is represented by the inner circle. (**A**) In darkness, phytochrome resides in the cytosol. PIF3 interacts with the DNA as negative transcription factor and inhibits transcription of certain genes. (**B**) In the light, phytochrome converts to Pfr, and this form interacts with FAR1 and FHY1. The complex moves into the nucleus (arrow) (**C**) This triggers transfer to the nucleus. PIF3 moves towards PhyA (**D**) FAR1 and FHY1 move back to the cytosol and phytochrome interacts with PIF3, thereby triggering its degradation possibly by phosphorylation and sumoylation. The green arrow indicates that specific transcription can start due to removal of PIF3 from the promotor. (**E**) PIF3 degrades, and new phytochrome in the Pr form appears in the cytosol. (**F**) PhyA interacts with SPA1 and with COP1, and PhyA is degraded.

**Figure 5 biomolecules-14-00009-f005:**
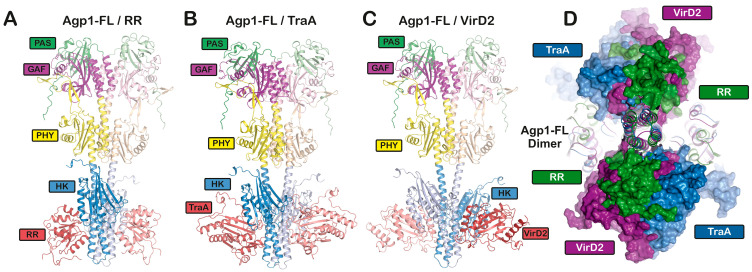
Alphafold 2.3.1 [15] models of Agp1 (Uniprot-ID Q7CY45) dimer in complex (**A**) with its response regulator; (**B**) with TraA (Uniprot ID Q44349, amino acids 1-243); and (**C**) with VirD2 (Uniprot-ID P18592). In all three cases, an interaction was modeled between both ATPase regions of the dimer Agp1. (**D**) Surface representation of a top view along the two-fold symmetry axis of the Agp1 dimer overlay. RR (in green), TraA (in blue) and VirD2 (in purple) interact according to the AF models at very similar interaction points of the two ATPases in the Agp1-FL dimer. The plDDT values are presented in false colors in the Appendix A. These give an impression of the prediction quality, which was high in all cases.

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
