# Peer review of "Phytochrome-Interacting Proteins"

_biomolecules, 2023, doi:10.3390/biom14010009_

Round 1

Reviewer 1 Report

Comments and Suggestions for Authors

The authors present a comprehensive, mainly structure-oriented review of phytochrome proteins and their interactions, as they have been studied so far in the literature. Most of the review focuses on plan phytochromes, however, fungal and prokaryotic members are also represented.

The review is, for the most part, well-written and comprehensive. However, several issues need to be addressed.

Major Points:

Point 1. The title, "Phytochrome Interacting Proteins" is too short, too generic, and does not represent the content of the review. The authors should provide a more detailed title for their manuscript, e.g., "Structural and functional aspects of phytochromes and their interacting partners" (just a coarse example of what the title could be).

Point 2: The manuscript flow jumps from Figure 1 (domain organization) to Figure 3 (Cryo EM structure and AlphaFold2 model). Figure 2 (the chromophore chemical structures) is never referenced in the paper, neither is Figure 4 (schematic representation of plant phytochrome activity). The authors need to make sure the flow of the manuscript is correct, and that all figure elements are properly presented and referenced.

Point 3: Figure 1 needs a better description/figure legend. None of the domain elements given is explained. Also, what is the meaning of the red line? Is it the binding site of the chromophores?

Point 4: Section 2 of the manuscript ("2. Interaction partners of plant phytochromes") needs to be expanded. First of all, the website the authors reference with a URL is not just a website; it is Biogrid, one of the most important biomolecular interaction databases. Make sure that it is properly referenced and that its publication is cited. Secondly, the statement "our selection of interacting proteins was rather based on intuition" is UNACCEPTABLE for a scientific publication. The authors should expand this section and provide their reasoning for selecting the interacting partners they present.

Point 5: A lot of the interactions described in sections 4-13 could be presented in a multi-panel figure; this would make the data presentation more comprehensive and improve the quality of the manuscript. Also, there is no need to provide all these evidence in separate sections, each of which is only one paragraph long; these could be merged in a single section.

Point 6: Since the authors have predicted their own models instead of using an existing model from AlphaFoldDB, more detail needs to be given on the procedure: which sequences were used (and their accession codes in a database, e.g. UniProt), which modeling scheme was selected (template-based, de novo?) etc. The modeling results should also be graphically presented in a manner that helps readers evaluate their quality, e.g. colored by pLDDT (which would help identify regions with low structural confidence), and accompanied by Predicted Aligned Error (PAE) graphs.

Point 7: The authors mention that the His-kinase orientation is significantly different between the available Cryo-EM structure and their AlphaFold prediction, but they never really present their thoughts on this discrepancy. Is the AlphaFold model inaccurate or is the CryoEM structure? If so, why? Is this difference somehow connected to the overall quality of the interdomain arms connecting the domains? (something that could be evaluated by examining the B-factor in the CryoEM structure and the pLDDT score in the AlphaFold models).  Could the different orientations of the domain influence binding interactions?

Minor: 

Point 1. Figure 2 can be reduced in size; there is no need for the 3 substances to take an entire page.

Point 2: In Figure 3, panel A, the His-kinase domain of PhyB is colored orange (and labeled HKRD). In panel B, the equivalent position of Agp1 is colored cyan and labeled HK. Please re-render the structures using the same color and label for the domain, for consistency.

Author Response

I thank all reviewers for their helpful comments. I aplogize that we have left some errors in the submitted manuscript, but am sure that the errorss are now reduced or eliminated and that the manusript is suitable for publication.

The authors present a comprehensive, mainly structure-oriented review of phytochrome proteins and their interactions, as they have been studied so far in the literature. Most of the review focuses on plan phytochromes, however, fungal and prokaryotic members are also represented.

The review is, for the most part, well-written and comprehensive. However, several issues need to be addressed.

Major Points:

Point 1. The title, "Phytochrome Interacting Proteins" is too short, too generic, and does not represent the content of the review. The authors should provide a more detailed title for their manuscript, e.g., "Structural and functional aspects of phytochromes and their interacting partners" (just a coarse example of what the title could be).

>REPLY We have written an article about phytochrome interacting proteins, this was agreed upont with the editors. We would like to leave the title as it is.

Point 2: The manuscript flow jumps from Figure 1 (domain organization) to Figure 3 (Cryo EM structure and AlphaFold2 model). Figure 2 (the chromophore chemical structures) is never referenced in the paper, neither is Figure 4 (schematic representation of plant phytochrome activity). The authors need to make sure the flow of the manuscript is correct, and that all figure elements are properly presented and referenced.

>REPLY: we have now improved figure legends. We have taken care that all abbreciations are explained. We have exchanged figures 1 and 2, because chromophores, now in figure 1, are mentioned earlier in the text. In the previous version we had forgotten to refer to the figure.

Point 3: Figure 1 needs a better description/figure legend. None of the domain elements given is explained. Also, what is the meaning of the red line? Is it the binding site of the chromophores?

>REPLY. There is now a long legend of figure 1, now figure 2. the red line is the chromophore binding site.

Point 4: Section 2 of the manuscript ("2. Interaction partners of plant phytochromes") needs to be expanded. First of all, the website the authors reference with a URL is not just a website; it is Biogrid, one of the most important biomolecular interaction databases. Make sure that it is properly referenced and that its publication is cited. Secondly, the statement "our selection of interacting proteins was rather based on intuition" is UNACCEPTABLE for a scientific publication. The authors should expand this section and provide their reasoning for selecting the interacting partners they present.

>REPLY: we have changed according to the reviewer. The URL remained but in addition the database is cited by the latest publication. „Based on intuition“ was reokaced by „Our selection below covers most of the database entries. “

Point 5: A lot of the interactions described in sections 4-13 could be presented in a multi-panel figure; this would make the data presentation more comprehensive and improve the quality of the manuscript. Also, there is no need to provide all these evidence in separate sections, each of which is only one paragraph long; these could be merged in a single section.

>REPLY: The PIF3 subchapter is a long subchapter, others are shorter. By the different titles we want to express that the interacting proteins can differ significantly from each other wiht respect to their functions and that alread many interactionn partners are known. It is necessary to have subtitles, becuase of the overall lengths of the section, but it is not possible to take out some subtitles, e.g. thos of short paragraphs, We have aa figure in which 5 interaccing proteins are shown.

Point 6: Since the authors have predicted their own models instead of using an existing model from AlphaFoldDB, more detail needs to be given on the procedure: which sequences were used (and their accession codes in a database, e.g. UniProt), which modeling scheme was selected (template-based, de novo?) etc. The modeling results should also be graphically presented in a manner that helps readers evaluate their quality, e.g. colored by pLDDT (which would help identify regions with low structural confidence), and accompanied by Predicted Aligned Error (PAE) graphs.

>REPLY: We have now given Uniprot ID in the legend of figure 6. The version number of Alphafold is given now (2.3.1), it was used with databases of May 2023. The program uses information from templates for learning, but calculations of model must be regarded as de novo. We made images presenting the pLDDT in color code. In figure 5 structures, colors are used for other information, so we show these as extra figures.

Point 7: The authors mention that the His-kinase orientation is significantly different between the available Cryo-EM structure and their AlphaFold prediction, but they never really present their thoughts on this discrepancy. Is the AlphaFold model inaccurate or is the CryoEM structure? If so, why? Is this difference somehow connected to the overall quality of the interdomain arms connecting the domains? (something that could be evaluated by examining the B-factor in the CryoEM structure and the pLDDT score in the AlphaFold models).  Could the different orientations of the domain influence binding interactions?

 >REPLY: We compared the 4 available Cryo-EM structures of plant phytochromes. They are morer or less identical. It means they are very likely correct and the Alphafold model is not correct. This is now mentioned in the new version.

Minor: 

Point 1. Figure 2 can be reduced in size; there is no need for the 3 substances to take an entire page.

 >REPLY: We made larger sizes but in the final version the figures must be smaller. We reduced the sizes in the manuscript, but I am not sure if not the publisher should make the right size.

Point 2: In Figure 3, panel A, the His-kinase domain of PhyB is colored orange (and labeled HKRD). In panel B, the equivalent position of Agp1 is colored cyan and labeled HK. Please re-render the structures using the same color and label for the domain, for consistency.

>REPLY. The histidine kinase of plant phytochromes has lost its kinase activity, several key amino acids are mutated via the standard HK sequence. Therefore it is often called HKRD. It is inconsistend with Fig. 2, where we name it HK. The text does not go into the discussion on HK activity and its evolution, it is a separate field.

Reviewer 2 Report

Comments and Suggestions for Authors

Phytochromes are pivotal photoreceptors governing physiological activities among a wide range of species including plants, bacteria and fungi. In special, the plant phytochromes have been intensively studied for over 70 years. Previous studies mainly focused on the photochemistry, spectral properties, structures of N-terminal photosensory modules and tremendous signaling components and physiological roles of phytochromes, and the predominant theoretical framework of phytochrome biology has been synthesized. However, molecular mechanisms of phytochrome signaling pathways based on the high resolution structures of full length phys and the ones in complex with their signaling partners remain largely elusive. Benefit from the advancements of cryo-EM techniques, these issues are ongoing dissected and progresses have been achieved recently. Hence, this review is very timely and provides valuable references for research peers and general readers to grasp the basic principles and latest progresses in phytochromes, especially their interacting partners. In general, the review is well-written and I have only several minor concerns to be addressed as follows.

Minor errors:

1. Line 18: AI for “artificial intelligence”

2. Line 34: should be “Pfr to Pr”?

3. Lines 43-44: PAS-GAF-PHY, spell out their full names.

4. Line 56: PCM, the full name.

5. Line 59-60: figure legends should be added in figure 1 and the abbreviations in the figure should be annotated.

6. Figure 2: no citation in the main text.

7. Line 63: “some cyanobacterial chromophores” better be replaced by “some algae”?

8. Figure 3: again, abbreviations should be spelled out.

9. Line 79: the other should be another?

10. Line 85: delete the second “of” and “are” should be “is”; “under phytochrome’s control” or “under the control of phytochrome”. Also in Line 87.

11. Line 89: “as the first step of the photocycle” should be moved to the end of this sentence.

12. Line 91: result in

13. Line 93: “activities of other enzymes”?; “and/or”

14. Line 95: phytochromes’ PAS-GAF-PHY tridomains

15. Line 100: reference 10 is not suitable here, adding the reference “Otero et al., Sci. Adv. 7, eabh1097 (2021)” would be better.

16. Line 105, reference 31 is not related to this paper and may be misused, and reference 26 can replace it.

17. Line 108: is the “Ipomea” right?

18. Line 139: PIF, full name

19. Line 156: remove “by”

20. Line 173: “et al.”

21. Line 177: necessary for “PIF3” binding

22. Line 187: remove the “of” or rephrase the sentence

23. Line 215: it should be “Fig. 4”, in addition, Figure # or Fig. # should follow the manuscript instructions.

24. Line 218: “cryptochrome (cry)”

25. Line 241: “the attention was drawn”

26. Line 284: reference 71 needs double checking.

27. Line 296: delete “3333”

28. Line 314: “chrome, the partner protein”

29. Line 323: “FHY1.” “nucleus.”

30. Line 325: the legends for E and F are missing. In addition, the title of this figure can further be clarified as “plant phytochrome A”.

31. Line 366: “at its C-terminus”, delete the second DrBphP.

32. Line 400: “TraA and VirD2”?

33. Line 436: “approaches.”

34. Line 439: “efficiency.”

35. Line 459: reaction regulator should be “response regulator”.

36. Line 480: reference 100 is the same to reference 98.

37. Line 489: “fewer” interaction partner?

38. Line 704: the title in this reference is incomplete.

39. Line 758: delete the reference 100.

Author Response

Phytochromes are pivotal photoreceptors governing physiological activities among a wide range of species including plants, bacteria and fungi. In special, the plant phytochromes have been intensively studied for over 70 years. Previous studies mainly focused on the photochemistry, spectral properties, structures of N-terminal photosensory modules and tremendous signaling components and physiological roles of phytochromes, and the predominant theoretical framework of phytochrome biology has been synthesized. However, molecular mechanisms of phytochrome signaling pathways based on the high resolution structures of full length phys and the ones in complex with their signaling partners remain largely elusive. Benefit from the advancements of cryo-EM techniques, these issues are ongoing dissected and progresses have been achieved recently. Hence, this review is very timely and provides valuable references for research peers and general readers to grasp the basic principles and latest progresses in phytochromes, especially their interacting partners. In general, the review is well-written and I have only several minor concerns to be addressed as follows.

Minor errors:

1. Line 18: AI for “artificial intelligence”

2. Line 34: should be “Pfr to Pr”?

>REPLY: The text around line 34 is : „Few phytochromes are comparable to this group but have a stable Pfr, i.e. the conversion from Pr to Pfr is only possible by light. A third group, the so-called bathy phytochromes, have a Pfr dark form [5, 6]. They are also synthesized in the Pr form, but by dark conversion, Pr converts to Pfr.“

3. Lines 43-44: PAS-GAF-PHY, spell out their full names.

>REPLY: It is now spelled out in line 48 and citations are given in the legend of figure 2

4. Line 56: PCM, the full name.

>REPLY This is given now

5. Line 59-60: figure legends should be added in figure 1 and the abbreviations in the figure should be annotated.

>REPLY: Figure legends were all completed and we have explained the abbreviations . Note that Figure 1 and 2 were exchanged according to appearance in the text

6. Figure 2: no citation in the main text.

>REPLY: It was forgotten in the first version. We have inserted the citation to the figure and as a consequence we had to exchange 1 and 2.

7. Line 63: “some cyanobacterial chromophores” better be replaced by “some algae”?

>REPLY. Ther was mistake in the legend. It is now „Chromophores of phytochromes. Phytochromobilin is the chromophore of plant phytochromes, phycocyanobilin is the chromophore of some cyanobacterial and green algal phytochromes and of cyanobacteriochromes, biliverdin is the chromophore of bacterial and fungal phytochromes.“ Algae are now also mentioned

8. Figure 3: again, abbreviations should be spelled out.

>REPLY: Agp1 is now given as Agrobacterium phytochrome Agp1. PDB has now a citation, it is usually not spelled out.

9. Line 79: the other should be another?

>REPLY: thank you, it was corrected

10. Line 85: delete the second “of” and “are” should be “is”; “under phytochrome’s control” or “under the control of phytochrome”. Also in Line 87.

>REPLY: This was modified accordingly, but the plural form are was kept

11. Line 89: “as the first step of the photocycle” should be moved to the end of this sentence.

>REPLY: was moved there

12. Line 91: result in

>REPLY: must be “results in” : was not changed

13. Line 93: “activities of other enzymes”?; “and/or”

>REPLY Now “These are translated into modulations of the histidine kinase and changes in its enzyme activities, in other enzymatic activities or in interactions with other proteins as the next step of signal transduction. “

14. Line 95: phytochromes’ PAS-GAF-PHY tridomains

15. Line 100: reference 10 is not suitable here, adding the reference “Otero et al., Sci. Adv. 7, eabh1097 (2021)” would be better.

16. Line 105, reference 31 is not related to this paper and may be misused, and reference 26 can replace it.

>REPLY: Thank you, that must have been the result of unnoted misinsertion. There are now the correct citations for Cph1 and Agp1

17. Line 108: is the “Ipomea” right?

>REPLY: yes, it is just not in the title

18. Line 139: PIF, full name

>REPLY: corrected and spelled out

19. Line 156: remove “by”

>REPLY: was removed

20. Line 173: “et al.”

>REPLY: was corrected

21. Line 177: necessary for “PIF3” binding

>REPLY: this wass serious typo, now corrected

22. Line 187: remove the “of” or rephrase the sentence

>REPLY: now “Through Western blot analysis they confirmed and expanded the data of Bauer et al. [54], who demonstrated that the immunochemically measurable native PIF3 protein decreases below detectable levels in wild-type Arabidopsis seedlings within 260 min of exposure to continuous red light. “

23. Line 215: it should be “Fig. 4”, in addition, Figure # or Fig. # should follow the manuscript instructions.

REPLY: was corrected , the rule is Figure, was correctedd too

24. Line 218: “cryptochrome (cry)”

REPLY: This was changed

25. Line 241: “the attention was drawn”

REPLY: this was changed

26. Line 284: reference 71 needs double checking.

REPLY: the citation was incorrect, a correct one was inserted

27. Line 296: delete “3333”

>REPLY: was deleted, probably by unwanted keystrikes just before submission

28. Line 314: “chrome, the partner protein”

>REPLY: comma entered

29. Line 323: “FHY1.” “nucleus.”

>REPLY: this is correct: FAR1 and FHY1 move back to the cytosol

30. Line 325: the legends for E and F are missing. In addition, the title of this figure can further be clarified as “plant phytochrome A”.

>REPLY: E, PIF3 degrades, new phytochrome in the Pr form appears in the cytosole. F, PhyA interacts with SPA1 and with COP1, PhyA is degraded. Phytochrome A: A is added

31. Line 366: “at its C-terminus”, delete the second DrBphP.

>REPLY: was changed

32. Line 400: “TraA and VirD2”?

>REPLY: yes, corrected

33. Line 436: “approaches.”

REPLY: full stop was added

34. Line 439: “efficiency.”

>REPLY: another full stop was missing and was added

35. Line 459: reaction regulator should be “response regulator”.

>REPLY: was corrected

36. Line 480: reference 100 is the same to reference 98.

>REPLY: it is now ref 15 at two positions, see comments above

37. Line 489: “fewer” interaction partner?

>REPLY: yes and corrected

38. Line 704: the title in this reference is incomplete.

>REPLY: corrected

39. Line 758: delete the reference 100.

>REPLY: this will be done by Endnote

Reviewer 3 Report

Comments and Suggestions for Authors

This manuscript summarizes the literatures on phytochromes and their interacting partners. The authors also use AlphaFold prediction to speculate on the potential interactions between bacterial phytochromes and their partners. I find this manuscript useful for people working in this field. A few statements can be improved as listed below.   

1. Please spell out all abbreviations when first used.

2. Line 55-56 “plant phytochrome structures have parallel histidine kinase regions……”, Because plant phytochromes have lost their histidine kinase activity, it is not accurate to call them histidine kinase regions. 

3. Figure 1. Label the last amino acid number for the four sketched proteins. Consider distinguishing the naming of C-terminal regions of Plant versus Bacterial phytochromes.

4. Figure 2 is not cited in the main text.

5. For Figure 3B, there is a full-length experimental structure of a bacterial phytochrome (PMID: 28275738). Please consider using experimental structure as example instead of the AlphaFold model as well as the PCM module determined by the authors themselves.  

6. Wrong citations in Line 100 “crystal structures of full-length phytochrome are known [10,27]”. Neither ref. 10 nor 27 reports crystal structure. 

7. Line 103 “the PAS-GAF-PHY tridomains are usually arranged in a parallel manner”. This statement may be fine for the truncated PAS-GAF-PHY tridomain itself and bacterial phytochromes. But the tridomains of plant AtPhyA and AtPhyB are antiparallel in their full-length structures. Based on sequence similarity among plant phytochromes, it is possible that all plant PAS-GAF-PHY tridomains adopts the antiparallel arrangement.   

8. Line 104-105 “antiparallel arrangements have been described for the cyanobacterial phytochrome Cph1 and the A. fabrum phytochrome Agp1 [25,31]” This statement is problematic. Ref. 31 is a wrong citation. The Cph1 crystal structure was firstly reported in PMID: 18799745. But more importantly, the native Agp1 (ref. 25) adopts parallel arrangement, and the antiparallel arrangement of the Agp1 PAS-GAF-PHY tridomain (ref. 25) was a result of engineered mutations that altered the interface property. The engineered Agp1 is not a convincing example for tridomain antiparallel arrangement.

9. Line 125-129. This part describes the arrangement of PhyA and PhyB and should cite ref 28-30.

10. Line 123-124: “The interaction between monomers may be the first interaction...”. Line 140-141 “The interaction between phytochrome and phytochrome interacting protein PIF3 was the first interaction…”. Confusing - which one is the first interaction?

11. Line 148: “C-terminal PAS-PAS-Histidine kinase…” Again, the C-terminal of plant phytochrome is not a histidine kinase.

12. Line 209-217 repeats line 197-203. 

13. Line 215: “see model in Fig. 3”. This should be Fig. 4 

14. Line 259 “(constitutively photomorphogenetic)” repeat of line 258

15. Line 269 “one PAS domain A of the C-terminus….” Typo?

16. Line 277 and line 278 “are” should be “is”

17. Line 323: “these triggers…” trigger.

18. Line 361-363: “In many bacterial phytochromes……is linked to the C-terminal part of the histidine kinase region.” State one or two examples.

19. Line 365-366: “A structural study of …. the interaction of response regulator and DrBphP at the C-terminus of DrBphP.” A transient interaction was observed only when the response regulator was covalently connected to the C-terminus of DrBphP via a 12-residue linker. This caveat should be noted. 

20. There are several uses of “Agrobacterium fabrum” and “A. fabrum”. For simplicity, please consider using only “A. fabrum” after the first use of full name. 

21. Line 411. there is no citation for AlphaFold where it appears the first time. But the same AlphaFold was cited twice later.

22. Line 445-446. “Moreover, the overall folding of PhyA and PhyB differed from the cryo-EM structures”. This is confusing, please rephrase.   

Comments on the Quality of English Language

No issue

Author Response

This manuscript summarizes the literatures on phytochromes and their interacting partners. The authors also use AlphaFold prediction to speculate on the potential interactions between bacterial phytochromes and their partners. I find this manuscript useful for people working in this field. A few statements can be improved as listed below.   

1. Please spell out all abbreviations when first used.

 >REPLY: This was done, see also comments to reviewer 2

2. Line 55-56 “plant phytochrome structures have parallel histidine kinase regions……”, Because plant phytochromes have lost their histidine kinase activity, it is not accurate to call them histidine kinase regions. 

>REPLY. The term histidine kinase like was used for plant phytochromes, where possible

3. Figure 1. Label the last amino acid number for the four sketched proteins. Consider distinguishing the naming of C-terminal regions of Plant versus Bacterial phytochromes.

 >REPLY: We have published with and without length information. I haave checked other articles, and there are usually no length informations in cartoons with domain arrangement. Also, we have not information about protein lengths in the article. In order to keep it simple, we would like to leave this figure, note it has switched to position 2.

4. Figure 2 is not cited in the main text.

 >REPLY: it is now cited, tee comments above

5. For Figure 3B, there is a full-length experimental structure of a bacterial phytochrome (PMID: 28275738). Please consider using experimental structure as example instead of the AlphaFold model as well as the PCM module determined by the authors themselves.  

 >REPLY: The published structure is incomplete. We know from several comparisons that the Alphafold model should be correct. The article is more about interaction than about structure . For interaction, the full length is required. See also Figure 5.

6. Wrong citations in Line 100 “crystal structures of full-length phytochrome are known [10,27]”. Neither ref. 10 nor 27 reports crystal structure. 

 >REPLY: Thank you. It was corrected, see also repla to reviewer 2

7. Line 103 “the PAS-GAF-PHY tridomains are usually arranged in a parallel manner”. This statement may be fine for the truncated PAS-GAF-PHY tridomain itself and bacterial phytochromes. But the tridomains of plant AtPhyA and AtPhyB are antiparallel in their full-length structures. Based on sequence similarity among plant phytochromes, it is possible that all plant PAS-GAF-PHY tridomains adopts the antiparallel arrangement.   

>REPLY: This is right. All four published plant phy structures are very simiilar. This is not indicated in the text.

8. Line 104-105 “antiparallel arrangements have been described for the cyanobacterial phytochrome Cph1 and the A. fabrum phytochrome Agp1 [25,31]” This statement is problematic. Ref. 31 is a wrong citation. The Cph1 crystal structure was firstly reported in PMID: 18799745. But more importantly, the native Agp1 (ref. 25) adopts parallel arrangement, and the antiparallel arrangement of the Agp1 PAS-GAF-PHY tridomain (ref. 25) was a result of engineered mutations that altered the interface property. The engineered Agp1 is not a convincing example for tridomain antiparallel arrangement.

 >REPLY: the citation was corrected. The orientation of subunits in the dimer could become interesting, because in plant phytochromes it is also an antiparallel arrangement. We take Cph1 and Agp1 as an example, even though the latter is a mutant. If we say, in morst cases parallel, in some cases antiparallel, I think we can also use a mutant. Agp2-PCM can also form antiparallel dimers. These are also mutants. Since we do not go into a deeper discussion on thiss topic, I think we can leave this Agp1 example.

9. Line 125-129. This part describes the arrangement of PhyA and PhyB and should cite ref 28-30.

>REPLY: this is corrected now

10. Line 123-124: “The interaction between monomers may be the first interaction...”. Line 140-141 “The interaction between phytochrome and phytochrome interacting protein PIF3 was the first interaction…”. Confusing - which one is the first interaction?

REPLY: Now: „The interaction between monomers may be regarded as a very stable protein interaction of phytochrome“ and „The interaction between phytochrome and phytochrome interacting factor 3 (PIF3) was the first interaction for a phytochrome with another protein to be discovered.“

11. Line 148: “C-terminal PAS-PAS-Histidine kinase…” Again, the C-terminal of plant phytochrome is not a histidine kinase.

REPLY: This was modified to histidine kinase like whereever necessary.

12. Line 209-217 repeats line 197-203. 

>REPLY: should be regarded as a summary of the whole subchapter

13. Line 215: “see model in Fig. 3”. This should be Fig. 4 

 >REPLY was changed

14. Line 259 “(constitutively photomorphogenetic)” repeat of line 258

 >REPLY: thank you, the second was deleted

15. Line 269 “one PAS domain A of the C-terminus….” Typo?

1>REPLY: we deleted „A“

6. Line 277 and line 278 “are” should be “is”

 >REPLY: thank you, was changed

17. Line 323: “these triggers…” trigger.

 >REPLY: changed to „this ttriggers“

18. Line 361-363: “In many bacterial phytochromes……is linked to the C-terminal part of the histidine kinase region.” State one or two examples.

 >REPLY: inserted: such as Agp2 and refer ti Figure 1 (old Figure 2) where the domain arrangements are shown

19. Line 365-366: “A structural study of …. the interaction of response regulator and DrBphP at the C-terminus of DrBphP.” A transient interaction was observed only when the response regulator was covalently connected to the C-terminus of DrBphP via a 12-residue linker. This caveat should be noted. 

 >REPLY: Experimental data od DrBphP are too incomplete. We just want to say that there are experimental data that are consistent with our AF model

20. There are several uses of “Agrobacterium fabrum” and “A. fabrum”. For simplicity, please consider using only “A. fabrum” after the first use of full name. 

 >REPLY:Agrobacterium was replaced by A. with 2 exceptions

21. Line 411. there is no citation for AlphaFold where it appears the first time. But the same AlphaFold was cited twice later.

>REPLY it is now cited and the version number is given

22. Line 445-446. “Moreover, the overall folding of PhyA and PhyB differed from the cryo-EM structures”. This is confusing, please rephrase.   

>REPLY: This was rephrased, see comment above

Round 2

Reviewer 1 Report

Comments and Suggestions for Authors

The authors have tried to address my concerns and in doing so, have improved the quality of the manuscript.

Please note that I detected a significant number of language errors that need to be corrected. However, these can be corrected during the proofing stage.

Comments on the Quality of English Language

Please note that I detected a significant number of language errors that need to be corrected. However, these can be corrected during the proofing stage.